# Feeding and Activity Environments for Infants and Toddlers in Childcare Centers and Family Childcare Homes in Southeastern New England

**DOI:** 10.3390/ijerph19159702

**Published:** 2022-08-06

**Authors:** Patricia Markham Risica, Jacqueline M. Karpowicz, Tayla von Ash, Kim M. Gans, Kristen Cooksey Stowers, Alison Tovar

**Affiliations:** 1Brown University School of Public Health, Providence, RI 02912, USA; 2Center for Health Promotion and Health Equity, Brown University School of Public Health, Providence, RI 02912, USA; 3Department of Behavioral and Social Sciences, Brown University School of Public Health, Providence, RI 02912, USA; 4Department of Human Development and Family Sciences, University of Connecticut, Storrs, CT 06269, USA; 5Institute for Collaboration in Health, Interventions and Policy, University of Connecticut, Storrs, CT 06269, USA; 6Department of Allied Health Sciences, University of Connecticut, Storrs, CT 06269, USA; 7Rudd Center for Food Policy and Health, University of Connecticut, Hartford, CT 06103, USA

**Keywords:** infant, toddler, childcare, feeding, nutrition, physical activity, screen-time

## Abstract

Few studies have documented the food and physical activity (PA) environments of childcare settings caring for children <24 months of age, although they may be key contributors to developing child PA and diet patterns. We used an adapted Environment and Policy Assessment and Observation tool to assess the food and activity environments for infants and toddlers in childcare centers (n = 21) and family childcare homes (FCCH) (n = 20) and explored differences by childcare type. Many similarities were found between childcare site types; however, centers used more recommended feeding practices than FCCH (e.g., 100% of center providers talked with toddlers about feelings of hunger or fullness compared to 18% of family childcare providers (FCCP), *p* < 0.01). Differences in non-recommended feeding practices (e.g., spoon feeding, bottle propping and encouraging unhealthy foods) were mixed between childcare types. Toddlers in centers spent more time playing at higher PA levels than those in FCCH (61 vs. 13 min, *p* < 0.001). Screen time was observed in FCCH, but not in centers. Differences between childcare types may indicate differential influences on infant and toddler feeding and PA behaviors, which could predict disparate obesity risk. Future research should further observe these behaviors in a larger sample of centers and FCCH to inform childcare interventions and policies.

## 1. Introduction

Early childhood is an important time for the development of diet and physical activity (PA) patterns that may contribute to obesity throughout life [1,2]. Approximately 47% of infants (birth to 1 year) and 54% of toddlers (1 to 2 years) spend at least one day in non-parental care [3]. Childcare attendance is significantly associated with an increased risk of obesity in cross-sectional research [4], as well as longitudinal studies showing higher later child obesity risk after exposure to childcare in the first year of life [5,6,7,8]. However, few studies have examined how the childcare environment impacts feeding behaviors and PA in children under age 2 [4,5,6,7,9,10,11,12,13,14,15,16].

Of all children in care by someone other than a relative, 67% are cared for in childcare centers, with 28% cared for by family childcare providers (FCCPs) [17]. These types of care differ because centers typically separate children into different groups or classrooms with different caregivers based on child age while FCCPs care for children of varying ages in their homes [18]. Thus, centers allow providers to tailor care by child age while FCCPs must simultaneously arrange meals and activities to accommodate children at different developmental stages [18]. FCCHs also tend to have less structured schedules and operate with more logistical and space constraints than centers [18]. 

National best-practice recommendations offer guidance to help ensure appropriate nutrition, PA and sedentary time for children in childcare [19,20,21]. Nutrition environments that adhere to best-practice recommendations in both centers [22] and FCCH [23] are associated with better diet quality intake among young children compared to diets of children in those that do not adhere to best-practice recommendations. Similarly, centers and homes that adhere to PA best-practice recommendations are associated with higher levels of PA among children in their care, compared to children in childcare settings not adhering to best practices [24,25,26]. 

Healthy child diet and activity are generally found for children in environments following best-practice recommendations, but the current literature mostly represents studies conducted with older preschool-aged children. Because the first 1000 days of life represent a critical period for the development and prevention of childhood obesity [27,28] further research is urgently needed to better understand the childcare environment for infants and toddlers. Furthermore, studies have not examined how adherence to best-practice recommendations differs across centers and FCCH, which may have important implications for interventions. Thus, the purpose of this study is to examine infant and toddler provider feeding and activity (including PA and screen time) practices in both childcare centers and FCCH and explore differences by type of childcare setting.

## 2. Materials and Methods

Recruitment for this study occurred in cities and towns in the Providence, Rhode Island (RI) area that have lower income levels and higher ethnic/racial diversity. Recruitment of FCCH included providers enrolled in the control group for the Healthy Start study [13,14], a randomized controlled trial of an intervention for FCCP caring for 2–5-year-old children in RI and Massachusetts (MA). In that study, FCCPs were recruited using direct mailings, phone calls to licensed FCCHs available in state databases and through state organizations overseeing Child and Adult Care Food Program (CACFP) trainings. Childcare centers were invited to participate from a state list of childcare centers in the Providence area. Arrangements were made for a trained observer to spend a full childcare day in the center/FCCH observing childcare provider practices and environments. Incentives of USD 30 were provided to the observed providers.

Demographic and other characteristics were surveyed from participating providers prior to observation [13] (Table 1). These data included gender, race, ethnicity, age, marital status, country of origin, years in the U.S., languages spoken at home, household income, household size, and participation in the following federal programs: Special Supplement for Women, Infants and Children (WIC), Supplemental Nutrition Assistance Program (SNAP), and CACFP. 

Provider behaviors and childcare environments were measured using an adapted version of the Environment and Policy Assessment and Observation tool (EPAO) that was created and pilot-tested to assess nutrition and PA environments in childcare for children ages 1.5 to 5 [29,30]. Observers for the Healthy Start study were extensively trained in conducting the EPAO in FCCH [13]. Training was developed for the new EPAO version for use with the same Healthy Start observers in FCCH and centers, and two graduate students. The adapted “Infant/Toddler EPAO” captures provider feeding, and activity practices, activity and screen time-related behaviors of children up to 24 months of age and the childcare physical environment [19,20,21,29,30,31,32]. Two age categories were observed separately due to the developmental differences between infants (ages 0–12 months) and toddlers (ages 13–24 months). The final tool measured the frequency of provider feeding (Table 2 and Table 3), PA and sedentary behavior (including screen time) practices (Table 4) and communications (Table 5). In this section, the terms “healthy foods,” and “unhealthy foods,” are used to indicate provider behaviors (e.g., encouraging, praising, or letting a child choose between) related to these food types. While we did not conduct a dietary assessment or specific food analysis, we followed the NAPSACC self-assessment classification of healthy foods which includes fruits, vegetables, potatoes that are not pre-fried, lower-fat and not pre-fried meats or meat alternatives, whole grain foods, water, and unflavored milk. Unhealthy foods include pre-fried potatoes or meats, high-fat meats or other foods, high-sugar high-fat foods, high-salt high-fat snacks, and sugary beverages. The observer conducted an assessment or asked the provider about what was being served to determine the content generally of a food if it was unclear.

EPAO Scoring

For 9 feeding items on topics of bottle and/or sippy cup use, overall atmosphere, and the use of food as reward or punishment, observers recorded a score of 0, 1 or 2+ based on the number of times (count) the practice was observed. Another 29 infant and 30 toddler feeding items on topics of provider food intake, sitting, conversation, interactions during feeding/meals; praise, reason with or talk to children about food; support/hinder self-regulation; spoon feeding; encouragement; allow multiple servings; make special food if a child refuses to eat; reward with non-food (activities); and make fruit easier to eat (e.g., peeling) were scored from 0 to 3:0 (never occurred), 1 (a little), 2 (sometimes), and 3 (a lot), sometimes with a 4 (not applicable) option. 

Scoring for PA and sedentary-related practices included timing (start/stop) for indoor, outdoor and sedentary activities with each coded for the level of activity. The observer indicated yes/no to whether indoor and outdoor equipment were present. These were then summed as a count of equipment available to stimulate activity (e.g., balls, tunnels, or slides) or sedentary behavior (e.g., infant gentle bounce seat, TV, VCR/DVD player, or computer). Additional items for both infants and toddlers described whether or not the child was exposed to TV or other media in general and during meals, and the visual environment (books, posters, website and written policies); whether a written summary of feeding and of activity was provided to parents at the end of the childcare day was indicated. 

Statistical analysis

Descriptive statistics (i.e., means/standard deviations, counts/percentages) were used to characterize sample demographics and provider practices. Observation count variables were dichotomized as occurred or did not occur for analysis as most were rarely observed more than once. Provider feeding behaviors and environments were separated into recommended practices (e.g., sitting with toddlers during a meal or supporting self-regulation) and non-recommended practices (e.g., using food as a reward or withholding PA as a punishment) in accordance with the Institute of Medicine, NAP SACC and National Academy of Sciences, Engineering and Medicine guidelines [19,20,21]. 

For activity variables, time and count constructs were assessed as continuous variables (i.e., time in active play, time in light play, time sedentary, number of provider-led activities, time spent outside, time with screens, number of pieces of equipment promoting PA, and number of pieces of equipment promoting sedentary behavior); the remaining items were dichotomized based on whether they occurred or not. Activities were coded from 1 (inactive) to 7 (vigorous), including: 1 inactive or sedentary (mostly sitting or lying as in a stroller), 2–4 light (slow crawling, tummy time, walking, and digging), 4–6 moderate (scooting, moderate crawling, walking fast, and marching), or vigorous 7 (fast crawling, cruising, and running).

Percentages were calculated as the proportion of sites in which each behavior or practice occurred. For these, the denominator only included observations where there was an opportunity to observe the behavior. For example, if no infants were observed being given a bottle, the site would not be included in the denominator for related practices as the opportunity for this observation did not occur. Chi-square analysis and Fisher’s exact test (for sample sizes less than 5) were used to analyze differences by site type for categorical variables, while *t*-tests were used for continuous variables with a *p*-value of less than 0.05 considered significant.

## 3. Results

We observed a total of 41 childcare sites: 20 FCCH and 21 centers, for an average of 6.4 h. In those sites, 87 infants (68 in 12 centers and 19 in 14 FCCH) and 100 toddlers (67 in 10 centers and 34 in 16 FCCH) were observed. Centers cared for an average of eight children per age-specific classroom. FCCPs cared for an average of 10 children overall. Only 14% of centers received CACFP compared to three quarters of FCCH (*p* < 0.01). Among all childcare sites, the lead provider was female, and averaged 12 years of experience working in childcare. The majority of providers were non-Hispanic white (59%); less than one-third (29%) were Hispanic; and most were married or living with a partner (63%). The highest level of educational attainment varied, with 39% of providers having a high school education and 31% having at least a college degree. Providers in FCCH were more likely to be of older age, Hispanic and born outside of the United States than providers in centers (*p* < 0.01 for all). No other demographics differed by site type (Table 1).

**Table 1 ijerph-19-09702-t001:** Sample Demographics and Other Characteristics, Overall and by Site Type.

Characteristic	All Sites n = 41	Centers n = 21	Family Childcare Homes n = 20	*p*-Value
** Site Characteristics **				
Hours observed (mean (sd))	6.4 (1.0)	6.4 (0.4)	6.45 (1.4)	0.818
Number of children in care (mean (sd))	8.6 (3.7)	7.67 (2.9)	9.6 (4.3)	0.107
Program receives food subsidies (n (%))	18 (44)	3 (14)	15 (75)	<0.001 *
** Provider Characteristics **				
Years working in childcare (mean (sd))	12.(8.8)	12.6 (9.0)	12.0 (8.8)	0.824
Female (n (%))	41 (100)	21 (100)	20 (100)	--
Age Group (n (%)) *				<0.001 *
≤34	11 (27.5)	11 (52.4)	0 (0)	
35–44	10 (25.0)	5 (23.8)	5 (26.3)	
45–54	13 (32.5)	4 (19.1)	9 (47.4)	
55–64	6 (15.0)	1 (4.8)	5 (26.3)	
Race/Ethnicity (n (%))				<0.001
Hispanic	12 (29.3)	1 (4.8)	11 (55.0)	
NH Black	3 (7.3)	3 (14.3)	0 (0)	
NH White	24 (58.5)	17 (81.0)	7 (35.0)	
NH Other	2 (4.9)	0 (0)	2 (10.0)	
Country of birth (n (%))				0.006 *
United States	29 (70.7)	19 (90.5)	10 (50.0)	
Other	12 (29.3)	2 (9.5)	10 (50.0)	
Marital status (n (%))				0.622
Married or living with partner	26 (63.4)	12 (57.1)	14 (70.0)	
Single, never married	11 (26.8)	7 (33.3)	4 (20.0)	
Divorced or widowed	4 (9.8)	2 (9.5)	2 (10.0)	
Highest level of education (n (%))				0.235
High school diploma or GED	16 (39.0)	6 (28.6)	10 (50.0)	
Associates degree or 60 semester credits	12 (29.3)	6 (28.6)	6 (30.0)	
Bachelor’s degree or higher	13 (31.7)	9 (42.9)	4 (20.0)	

* Age group is missing one value for FCCP. Race/Ethnicity other category includes 1 American Indian/Alaskan Native, 1 multiracial; Country of birth other category includes 6 born in Caribbean, 2 in South America, 1 in Central America, 2 in Cape Verde and 1 not specified; *: statistically significant difference.


*Recommended Feeding Practices*


Several recommended feeding practices were observed at somewhat similar frequencies in both centers and FCCH (Table 2). However, significantly more recommended feeding practices of both infants and toddlers were observed in centers compared with FCCH. For infants, this included: enthusiastically role modeling eating healthy food (58 centers vs. 14% FCCH, *p* = 0.032), praising infants for eating healthy foods (92 vs. 14%, *p* < 0.001), sitting with infants during a meal (92 vs. 43% *p* = 0.002), talking with infants about foods they were eating (100 vs. 43%, *p* = 0.002), encouraging (not forced) infants to try new foods (83 vs. 43%, *p* = 0.048) and talking about feelings of hunger and fullness (100 vs. 7%, *p* < 0.001). In contrast, center providers (58%) were less likely than FCCP (64%) to praise an infant for trying new or less preferred foods; *p* = 0.011.

**Table 2 ijerph-19-09702-t002:** Frequency of Recommended Feeding Best Practices and Provider–Child Interactions in Centers and Family Childcare Homes.

Provider Practices		Age Group	All Sites n = 52 (26 Infant, 26 Toddler) Observations	Centers n = 21 (12 Infant, 9 Toddler) Observations	Family Childcare Homes n = 20 (14 Infant, 17 Toddler) Observations)	*p*-Value
	Variable Type/Scoring		n (%)	n (%)	n (%)	
The provider enthusiastically role-modeled eating healthy foods.	Frequency None–A Lot	Infants	9 (35)	7 (58)	2 (14)	0.032 *
		Toddlers	7 (27)	4 (44)	3 (18)	0.093
The provider praised a child for trying new or less preferred foods.	Frequency None–A Lot	Infants	16 (62)	7 (58)	9 (64)	0.011 *
		Toddlers	10 (39)	6 (67)	4 (24)	0.028 *
The provider praised a child for trying healthy foods.	Frequency None–A Lot	Infants	13 (50)	11 (92)	2 (14)	0.000 *
		Toddlers	12 (46)	7 (78)	5 (29)	0.012 *
The provider let a child choose between two healthy food options.	Frequency None–A Lot	Infants	7 (27)	3 (33)	4 (29)	0.939
		Toddlers	16 (62)	7 (77)	9 (53)	0.093
The provider made fruits and vegetables easier to eat.	Frequency None–A Lot	Infants	14 (54)	8 (75)	6 (43)	0.135
		Toddlers	25 (96)	9 (100)	16 (94)	0.484
The provider sat with the children during the meal.	Frequency None–A Lot	Infants	17 (65)	11 (92)	6 (43)	0.002 *
		Toddlers	12 (46)	6 (67)	6 (35)	0.025 *
The provider talked with the children about the foods they were eating.	Frequency None–A Lot	Infants	18 (69)	12 (100)	6 (43)	0.002 *
		Toddlers	19 (73)	9 (100)	10 (59)	0.032 *
The provider talked with the children informally about nutrition.	Frequency None–A Lot	Infants	5 (19)	4 (44)	1 (7)	0.070
		Toddlers	14 (54)	9 (100)	5 (29)	0.001 *
The provider encouraged (not forced) children to try foods on their plates.	Frequency None–A Lot	Infants	16 (62)	10 (83)	6 (43)	0.048 *
		Toddlers	17 (66)	7 (77)	10 (59)	0.152
The provider reasoned with a child to eat.	Frequency None–A Lot	Infants	10 (38)	4 (33)	6 (43)	0.742
		Toddlers	23 (88)	9 (100)	14 (82)	0.205
Children in the classroom were observed feeding themselves.	0, 1, 2+ Count	Infants	18 (69)	12 (100)	6 (55)	0.008 *
		Toddlers	22 (85)	9 (100)	13 (93)	0.444
The provider talked about feelings of hunger or fullness with children.	Frequency None–A Lot	Infants	13 (50)	12 (100)	1 (7)	0.000 *
		Toddlers	12 (46)	9 (100)	3 (18)	0.000 *
Second helpings were served only after a child requested seconds and the provider asked the infant if he/she was still hungry.	Frequency None–A Lot	Infants	22 (85)	10 (83)	12 (86)	0.792
		Toddlers	21 (81)	9 (100)	12 (71)	0.086
When a child ate less than half of a meal or snack, the provider asked the child if he/she was full before removing the plate.	Frequency None–A Lot	Infants	19 (73)	10 (83)	9 (64)	0.315
		Toddlers	21 (81)	9 (100)	12 (71)	0.086

Obs.: observations; counts represent the number of centers/homes in which each practice was observed, percent represents the proportion of centers/homes that were observed using the behavior: * = *p* < 0.05.

Similarly, for toddlers, many recommended practices were common among centers, but were observed less frequently among FCCPs, including: praising a toddler for trying new or less preferred food (67 centers vs. 24% FCCH, *p* = 0.028), praising a toddler for trying a healthy food (78 vs. 29%, *p* = 0.012), sitting with toddlers during a meal (67 vs. 35%, *p* = 0.025), talking with toddlers about foods they were eating (100 vs. 59%, *p* = 0.032), talking with toddlers informally about nutrition (100 vs. 29%, *p* = 0.001), praising toddlers for eating healthy foods (78 vs. 29%), and talking about feelings of hunger and fullness (100 vs. 18%, *p* < 0.001).


*Non-recommended feeding practices*


Non-recommended feeding practices were rarely observed for providers in both centers and FCCH, although when observed, these practices were more commonly observed in centers than FCCH (Table 3). For example, providers in centers were more likely than those in FCCH to: use a bottle to soothe before trying anything else (46% centers vs. 0% FCCH, *p* = 0.005), praise an infant for cleaning their plate (92% vs. 50%, *p* = 0.030), provide second helpings to infants even when not asked for (25% vs. 0%, *p* = 0.037) and spoon-feeding to get infants to eat (100% vs. 0%, *p* = 0.053). Providers in centers were also more likely than those in FCCH to insist that a toddler eat a certain food (100% vs. 24%, *p* < 0.001). 

However, providers in centers were less likely than those in FCCH to force an infant to finish a bottle (0% vs. 36%, *p* = 0.022), remove the plate when an infant ate less than half of the meal without asking if he/she was full (17% vs. 64%, *p* = 0.006), or praise or compliment a toddler for eating an unhealthy food (11% vs. 77%, *p* = 0.003).


*Physical Activity Environment and Practices*


Infants averaged 6.5 min in active play, compared with 30 min for toddlers (Table 4). Toddlers in childcare centers were observed spending more time in active play than toddlers in FCCH (61.7 vs. 13.4 min, *p* = 0.003). Light play observed among both site types averaged 112.9 and 101.9 min for infants and toddlers, respectively. Infants in centers were observed spending more time in light play than those in FCCH (165.5 vs. 67.6 min, *p* < 0.001), but toddler light play did not differ between site type. Sedentary time averaged 49.5 and 28.2 min for infants and toddlers, respectively, and did not differ significantly between centers versus FCCH for either age group. The observed provider-led physical activities averaged 3.8 activities for infants and 4.3 for toddlers. Providers in centers were observed leading more activities than those in FCCH for both infants (6.4 vs. 1.5, *p* < 0.001) and toddlers (8.4 vs. 2.1, *p* < 0.001). No significant differences in outside time or play opportunities, infant-designated space or equipment were found between centers and FCCH.

**Table 3 ijerph-19-09702-t003:** Frequency of Non-Recommended Infant and Toddler Feeding Practices in Centers and Family Childcare Homes.

Provider Practices		Age Group	All Sites n = 46 (23 Infant, 23 Toddler) Observation)	Centers (n = 21: 12 Infant, 9 Toddler) Observations	Family Childcare Homes (n = 20: 11 Infant, 14 Toddler) Observations	
	Variable Type/Scoring		n (%)	n (%)	n (%)	*p* Value
A teacher forced a child to finish the bottle.	0, 1, 2+ Count	Infants	4 (18)	0 (0)	4 (36)	0.022 *
A teacher added cereal to a child’s bottle.	0, 1, 2+ Count	Infants	2 (8)	0 (0)	2 (18)	0.104
		Toddlers	1 (4)	0 (0)	1 (7)	0.429
A teacher propped a child’s bottle.	0, 1, 2+ Count	Infants	14 (60)	7 (58)	7 (63)	0.821
A teacher let a child fall asleep with a bottle.	0, 1, 2+ Count	Infants	7 (30)	4 (36)	3 (26)	0.476
		Toddlers	2 (8)	0 (0)	2 (14)	0.269
A teacher used the bottle (or food) to soothe a child before trying anything else.	0, 1, 2+ Count	Infants	5 (22)	5 (46)	0 (0)	0.005 *
		Toddlers	1 (4)	1 (11)	0 (0)	0.174
Child was allowed to walk around with their bottle or sippy cup.	0, 1, 2+ Count	Toddlers	4 (18)	1 (11)	3 (21)	0.743
Food (healthy or unhealthy) was used as a reward for good behavior.	0, 1, 2+ Count	Toddlers	1 (4)	1 (11)	0 (0)	0.161
Food (healthy or unhealthy) was taken away as punishment for misbehavior.	0, 1, 2+ Count	Toddlers	0 (0)	0 (0)	0 (0)	--
The provider praised/complimented a child for eating unhealthy foods.	Frequency None–A Lot	Infants	1 (4)	1 (8)	0 (0)	0.569
		Toddlers	14 (60)	1 (11)	13 (77)	0.003 *
The provider praised a child for cleaning his/her plate.	Frequency None–A Lot	Infants	18 (78)	11 (92)	7 (50)	0.030 *
		Toddlers	21 (92)	9 (100)	12 (71)	0.086
The provider pressured a child to eat more than they seemed to want.	Frequency None–A Lot	Infants	9 (40)	5 (42)	4 (29)	0.383
		Toddlers	4 (18)	1 (11)	3 (18)	0.743
Second helpings were served to a child even when the child did not ask for more.	Frequency None–A Lot	Infants	3 (14)	3 (25)	0 (0)	0.037
		Toddlers	7 (30)	3 (33)	4 (24)	0.486
When a child ate less than half of a meal or snack, the provider removed the plate without asking if the child was full.	Frequency None–A Lot	Infants	11 (48)	2 (17)	9 (64)	0.006 *
		Toddlers	6 (26)	4 (44)	2 (12)	0.398
The provider spoon-fed a child to get them to eat.	Frequency None–A Lot	Infants	12 (52)	12 (100)	0 (0)	0.053
		Toddlers	10 (44)	3 (33)	7 (41)	0.227
The provider insisted that a child eat a certain food.	Frequency None–A Lot	Infants	18 (78)	10 (83)	8 (57)	0.098
		Toddlers	13 (54)	9 (100)	4 (24)	0.000 *
The provider prompted a child to finish one food in order to receive another food or seconds of another food.	Frequency None–A Lot	Infants	1 (4)	1 (8)	0 (0)	0.353
		Toddlers	17 (74)	7 (78)	10 (59)	0.152
The provider allowed a child to have or take multiple servings of a food when more than one food or a large amount of food remains on the plate.	Frequency None–A Lot	Infants	16 (70)	5 (42)	11 (79)	0.087
		Toddlers	20 (86)	7 (78)	13 (76)	0.936
The provider made special allowances to provide something different from what has already been served for a child that refuses to eat.	Frequency None–A Lot	Infants	3 (14)	2 (17)	1 (7)	0.931
		Toddlers	1 (4)	0 (0)	1 (6)	0.273

* = *p* < 0.05; Obs.: observations; counts represent the number of centers/homes in which each practice was observed, percent represents the proportion of centers/homes that were observed using the behavior; *: statistically significant difference.


*Screen Time Environment and Practices*


Almost all FCCH had a television present, compared to less than a third of centers (*p* < 0.001 for both age groups). All sites in which children were observed having screen time were FCCH; infants at 7% of FCCH and toddlers at 12% of FCCH were observed having multiple episodes of screen time. Infants and toddlers were observed in screen time at FCCH for an average of 22.3 and 23.3 min, respectively, over the entire observation period compared with no screen time observed in centers (NS).

**Table 4 ijerph-19-09702-t004:** Time Spent by Level of Physical Activity and Screen time and Environmental Features Provided Overall and By Site Type.

Characteristic		Age Group	All Sites n = 52 (26 Infant, 26 Toddler) Observations	Centers n = 21 (12 Infant, 9 Toddler) Observations	Family Childcare Homes n = 20 (14 Infant, 17 Toddler) Observations	*p*-Value
	Variable Type/Scoring					
Time in active play (mean min. (sd))	Start/Stop (coded level)	Infant	6.5 (17.2)	10.7 (22.7)	2.9 (10.1)	0.256
		Toddler	30.1 (42.1)	61.7 (38.4)	13.4 (34.3)	0.003 *
Time in light play (mean min. (sd))	Start/Stop (coded level)	Infant	112.8 (67.2)	165.50 (46.8)	67.6 (45.3)	<0.001 *
		Toddler	101.9 (54.2)	97.1 (29.3)	104.4 (64.4)	0.753
Time sedentary (mean min. (sd))	Start/Stop (coded level)	Infant	49.5 (52.2)	49.8 (39.9)	49.2 (62.4)	0.977
		Toddler	28.2 (38.5)	29.6 (41.2)	27.5 (38.2)	0.899
Number of provider-led activities (number (sd))	Count	Infant	3.8 (2.9)	6.4 (1.5)	1.5 (1.5)	<0.001 *
		Toddler	4.3 (3.9)	8.4 (2.9)	2.1 (2.1)	<0.001 *
**Outside Time and Equipment**						
Had any outside time (n (%))	Yes/No	Infant	9 (35)	2 (17)	7 (50)	0.110
		Toddler	17 (65)	5 (56)	12 (71)	0.443
>1 outdoor play session (n (%))	Count Recoded Yes/No	Infant	1 (4)	0 (0)	1 (7)	1.000
		Toddler	6 (23)	1 (11)	5 (29)	0.380
Time spent outside (mean min. (sd))	Start/Stop	Infant	17.7 (28.4)	6.9 (18.2)	27.0 (32.6)	0.071
		Toddler	39.8 (44.5)	27.7 (31.7)	46.2 (49.6)	0.322
Designated play space outside apart from older children (n (%))	Yes/No	Infant	4 (24)	2 (18)	2 (33)	0.584
		Toddler	3 (14)	3 (33)	0 (0)	0.063
On-site garden present (n (%))	Yes/No	Infant	6 (27)	2 (17)	4 (40)	0.348
		Toddler	8 (35)	2 (22)	6 (43)	0.400
Pieces of equipment promoting physical activity (number (sd))	Count	Infant	8.4 (5.5)	8.8 (4.9)	8.1 (6.1)	0.759
		Toddler	10.0 (5.6)	12.9 (4.8)	8.5 (5.5)	0.055
Sedentary and Screen Time						
Had any screen time (n (%))	Yes/No	Infant	5 (19)	0 (0)	5 (36)	0.042 *
		Toddler	6 (23)	0 (0)	6 (35)	0.063
>1 episode of screen time (n (%))	Yes/No	Infant	1 (4)	0 (0)	1 (7)	1.000
		Toddler	2 (8)	0 (0)	2 (12)	0.529
Time with screens (mean min. (sd))	Start/Stop	Infant	12.0 (42.0)	0.0 (0.0)	22.3 (56.0)	0.182
		Toddler	15.2 (45.2)	0.0 (0.0)	23.3 (45.2)	0.218
Television present (n (%))	Yes/No	Infant	18 (69)	4 (33)	14 (100)	<0.001 *
		Toddler	18 (69)	2 (22)	16 (94)	<0.001 *
Pieces of equipment promoting sedentary behavior (number (sd))	Count	Infant	2.3 (1.6)	2.5 (1.6)	2.1 (1.7)	0.586
		Toddler	1.9 (1.6)	1.1 (1.5)	2.2 (1.5)	0.087

Designated play space missing for 9 infants and 5 toddler obs., on-site garden missing for 4 infant and 3 toddler obs., whether or not a summary of child activity was provided to parents was missing or not observed for 18 infant and 17 toddler obs; * = *p* < 0.05.


*Nutrition and Physical Activity Communications*


Best-practice recommendations including having written materials, books, posters or written policies/contracts about nutrition breastfeeding and PA in centers and FCCH are summarized in Table 5. None of the centers and 10% of FCCHs were observed to have materials about breastfeeding (NS). One-third of centers and no FCCHs were observed to have materials about bottle feeding (*p* < 0.001). Close to half of the centers provided materials on infant (43%) and toddler (48%) nutrition compared to far fewer FCCHs (25% and 5%, respectively, *p* < 0.001 for both). Taken together, 81% of centers and 30% of FCCHs had any written materials about food or nutrition available (*p* < 0.001). Summaries of feedings were provided by some centers to parents of infants (57%) and toddlers (29%) but in FCCHs were provided less often to parents of infants (0%) and toddlers (5%) (*p* < 0.05 for both).

Communication materials about PA were available for families with infants in 19% and families with toddlers in 24% of centers compared to no information on PA for either age group in FCCHs (*p* = 0.11 for infants and 0.05 for toddlers). A written summary of child activity was provided by centers to parents of infants at seven sites and to parents of toddlers in four sites, compared to no FCCHs providing infant activity summaries (*p* = 0.13) and only one providing toddler activity summaries (*p* = 0.05).

**Table 5 ijerph-19-09702-t005:** Communication materials visible in childcare and summaries available for parents.

Type	Age Group	Overall	Centers	Family Childcare Homes	*p* Value
		n (%)	n (%)	n (%)	
Written materials about nutrition available	Infant	14 (34)	9 (43)	5 (25)	<0.001
	Toddler	11 (27)	10 (48)	1 (5)	<0.001
Written materials about breastfeeding available		2 (5)	0 (0)	2 (10)	0.5
Written materials about bottle feeding available		7 (15)	7 (33)	0 (0)	<0.001
Written materials about any feeding available		23/41	17 (81)	6 (30)	<0.001
Written summary of feeding provided to parents	Infants	12 (29)	12 (57)	0 (0)	0.001
	Toddler	7 (17)	6 (29)	1 (5)	0.04
Written materials about PA available (n (%))	Infant	4 (10)	4 (19)	0 (0)	0.107
	Toddler	5 (12)	5 (24)	0 (0)	0.048
Written summary of child activity provided to parents (n (%))	Infant	7 (88)	7 (100)	0 (0)	0.125
	Toddler	5 (56)	4 (100)	1 (20)	0.048

## 4. Discussion

This study found several differences in observed nutrition and PA-related childcare provider practices between centers and FCCHs caring for infants and toddlers. Centers and FCCHs used many similar recommended feeding practices, but overall, more recommended and not-recommended feeding practices were observed in centers than in FCCHs. PA time was greater for toddlers in centers than in FCCHs, as was teacher/provider-led activities for both infants and toddlers. More TV exposure and time in sedentary behaviors were found in FCCHs compared with centers. Communications to parents regarding nutrition and PA were greater in centers than in FCCHs. 

In both settings, childcare providers engaged in several recommended practices for feeding, including: being responsive to infant and toddler satiety cues, establishing a pleasant feeding environment, role-modelling healthy eating, and encouraging self-feeding and self-regulation. However, fewer recommended feeding practices were observed in FCCH as compared to centers, which is similar to previous studies [33,34,35,36,37]. For example, other studies [33,35,36,37] found, as we did, that most providers sit with children while they eat, but finding more of this practice in centers than FCCP, or differences in this practice between these site types at all, have not previously been reported, especially among infants and toddlers. 

Our finding that infants and toddlers are generally not being required to “finish their plate” is somewhat in contrast to other studies where requiring preschool children in childcare to finish their plate and requesting children to eat certain foods have been reported frequently [16,38,39,40]. Providers are possibly not using these practices because they perceive infants and toddlers to be different than preschoolers. The findings that providers are not using food rewards is consistent with other literature among childcare providers of infants and toddlers [33], and preschoolers [16,40]. Trainings about best-practice feeding recommendations and guidelines might have made this recommendation clear to providers and it may be easier to implement than other feeding practices such as insisting on a clean plate. 

Non-recommended bottle- and spoon-feeding practices were also observed with varying frequency. Bottle propping was found in ~60% of observations in both settings. Bottle propping as well as more forced bottle-feeding was observed more frequently among FCCHs, while use of a bottle to soothe was observed more among centers. Although few studies on these behaviors have been conducted in childcare settings, bottle propping [33,41,42] has been found to be somewhat common (20–50%) in studies of lower-income and racial/ethnically diverse samples of parents, but still less frequently than in our observations. Center providers commonly encouraged more infant food intake through spoon feeding, although over one-third of providers in both settings were also observed to spoon-feed toddlers. Practices such as these are discouraged because they may decrease an infant’s reliance on hunger and satiety cues [43], which have been associated with a risk of overfeeding and childhood obesity [43,44,45,46], although we found no highlights of these practices in other childcare literature. 

Infants averaged less than ten minutes in active play across site types in our observations. Caring for our Children recommends for infants bouts of 3–5 min of tummy time, with increasing duration as they show that they are enjoying it [47]. Toddlers averaged 30 min of active play across site types, with much higher time in centers (62 min) compared to FCCHs (13 min, *p* = 0.003). The WHO recommends 180 min per day in a variety of types of PA for toddlers 1–2 years of age [48], and NAPSACC recommends 90 min or more of combined indoor and outdoor PA in childcare [49], although neither of these guidelines specify recommendations for active play time for children under 2 years of age. Caring for our Children recommends 60–90 min of moderate to vigorous play during an 8 h childcare day [47]. A systematic review of studies measuring PA levels among preschoolers in home-based childcare using accelerometry [50] found that moderate to vigorous play averaged 1.8 to 9.7 min per day and total PA averaged 10–34 min, which is similar to our findings. Examining the proportions of time spent in active play in our study, toddlers averaged 19% of measured time in active play (33% centers and 9% FCCHs) with 4% active time for infants, which was similar between centers and FCCHs. Other studies in FCCHs that used accelerometry found that preschool children averaged 10% active time [51,52]. 

The current study also found an average of 49 min of sedentary time during the observation time for infants and 29 min for toddlers, with no site differences. In contrast, a systematic review of studies measuring sedentary behaviors of 2–5 year-olds in home-based childcare found a much higher average of accelerometry-measured sedentary time (40–50 min) [50]. In our observations, infants averaged 29% and toddlers 18% of time as sedentary time in contrast to other studies in FCCHs using accelerometry with 2–5 year-old children that found 50% [51] and 63% [52] sedentary time. Differences in these findings may be due to the EPAO methods of observing time spent used in this study compared with the more technically accurate methodology of assessing actigraphy levels, but these differences might also be due to the age of children being observed (preschoolers vs. infants and toddlers). 

Providers in previous studies perceived young children to be sufficiently active on their own and, thus, saw little need to intentionally plan PA [53,54]. Empirical evidence of the specific levels of current PA and sedentary behavior among infants and toddlers in childcare is limited; however, studies with older children have found high amounts of time in sedentary behavior and low time in active PA in childcare settings [52,55,56,57,58]. It is clear that lower PA and higher sedentary behaviors are associated with higher risk of later cardiovascular and other health outcomes [59,60]. 

We found that providers led activities that got both infants and toddlers moving, although providers in centers led many more activities than those in FCCH. Teacher-led activities in our study averaged 4.3 min for toddlers and 3.8 min for infants, compared with 15 min of FCCH provider-led activity for 2–5-year-old children in FCCH in the Keys study [51]. Just over 85% of providers in another study were observed leading a planned PA class once or more per week among FCCH caring for 2–5-year-old children [15]. Differences in staff availability and training to lead physical activities might account for some of the differences between centers and FCCH [24]. 

Sedentary behavior can include quiet time of various sorts, but might also include screen time, which was assessed separately in this study. Infant and toddler observations averaged 22–23 min of screen time in FCCH, but no time with screens was observed in centers for either toddlers or infants. The observation of far less screen time in centers than in FCCHs is supported by other research [61]. The high screen time found in some FCCHs is particularly concerning in comparison to the Caring for our Children guidelines, which recommend no screen time in early childcare settings [47]. NAPSACC [49] and WHO [48] also recommend rare or no screen time for infants and toddlers. 

Differences seen between centers and FCCH in amounts and types of activity may also be due to the social, physical and policy environments, and differences in child age ranges. Center environments offer more PA opportunities and portable and fixed play equipment, which have been shown to promote higher PA and less sedentary time than FCCH [24]. Duration of outdoor play and characteristics of the indoor play space have been associated with increased MVPA [55,62,63]. However, associations between types of indoor and outdoor play equipment available with PA levels have been mixed [54,62]. While we found a near-significant difference in pieces of equipment for toddlers by site type (with centers having more pieces of activity promoting and fewer pieces of sedentary-promoting equipment than FCCH), it is unlikely that equipment alone resulted in the difference in active play that we observed across by site type. Barriers to PA for FCCP identified in other research include lack of space, including suitable indoor space, training, time, policies, encouragement by providers and concerns surrounding child injury and safety [37,53,64,65]. It is also difficult to engage children in a way that suitably encourages PA for the wide age range of children present in FCCH [66,67]. 

This study found room to increase adherence to best practices for feeding, PA, sedentary behaviors and screen time practices among both centers and FCCHs. Future interventions might consider the unique resources and demands of each site type in designing interventions to help providers meet feeding best-practices and activity guidelines for infants and toddlers. Studies to identify facilitators and barriers associated with providers meeting recommended guidelines would also provide the foundation for successful interventions. Non-recommended feeding practices were prevalent for both centers and FCCHs. Interventions and policies that target reducing these practices might be more successful if they include components to increase facilitators and reduce barriers to the preferred practices, particularly for FCCHs. Future research could also explore how CACFP standards, state licensing regulations, quality-rating-improvement systems, and wellness policies can be leveraged to better support recommended feeding and activity practices and reduce non-recommended practices within centers and FCCHs [68,69]. Additionally, the measurement of child food intake and activity levels would be an important addition to future the observation and interpretation of childcare provider practices.

The childcare environment is extremely important for the development of infant and toddler eating and activity behaviors. However, parents and the home environment are also very important and previous research has highlighted this influence on infant and child early eating [70,71,72,73,74,75,76] and PA behaviors [77,78,79], and overall obesity risk [76,80,81,82,83]. Future research should explore both of these environments together and identify opportunities to create more consistent healthy practices across these environments [37,84,85,86,87]. 

While considering these important findings and implications, limitations in this study must also be considered. First, the large number of comparisons made may have led to finding significant differences due to chance alone, so over-interpretation of any particular finding should be avoided. It is also possible that some of the differences observed were, at least in part, due to provider and child demographic differences between centers and FCCHs. However, demographic factors are unlikely to fully explain the marked differences observed in provider practices by site type. On the contrary, the modest sample size coupled with some missing data may have limited our power to detect some differences by site type. Additionally, observations of these small numbers of centers and FCCH in only Rhode Island and Massachusetts limit how generalizable these data are to a broader population. Furthermore, only one day was observed in each center or FCCH, which may not be representative of a provider’s regular behaviors, although for this reason, behaviors were averaged over the groups. 

Despite these limitations, to our knowledge, this has been the first study that uses direct observation to evaluate the environments of childcare centers and FCCHs for children under age two. While previous research has looked at parental feeding practices with infants [19,88,89,90] or childcare provider practices in regard to toddlers and preschoolers [13,16,22,38,39,40,51,61,91,92,93,94,95,96,97], the current study is a novel approach to examining the practices of childcare providers caring for children under two years, especially in FCCHs, which have been largely overlooked [24,33,62]. Our successful use of direct observation with the EPAO in both centers and FCCH to study infants and toddlers is consistent with multiple studies that have documented the accuracy and reliability of this method for preschoolers [36,51,98].

## 5. Conclusions

This study found differences in provider–child interactions between childcare centers and FCCH suggesting that the type of childcare may differentially influence providers’ infant and toddler feeding and activity behaviors. Because young children are so easily influenced by their environments, childcare environments have the potential to influence the development of infants’ and toddlers’ eating and PA behaviors and their subsequent risk of developing obesity. Interventions to improve childcare providers’ infant and toddler feeding and activity practices are very limited. For interventions to be successful, however, childcare environments need to be adequately supported. Although differences may be found between childcare types, considerable variety in organization and policies may exist with each childcare type, and deserves more scrutiny in a larger study design.

With a large proportion of children in childcare [99,100], additional research should target childcare factors that contribute to the risk of infant overweight and obesity. New studies with larger sample sizes and longer periods of observation are necessary to confirm areas of need that might require stronger support. Indeed, strengthened nutrition, PA, sedentary and screen-time policies and state licensing regulations may be needed to achieve better nutrition and PA practices in both types of care. These might include specific federal practices added in the United States to the US Department of Agriculture (USDA)-administered CACFP, the US Health and Human Services (HHS)-administered Quality Rating and Improvement System (QRIS), as well as food and activity sections of state and local licensing requirements. More robust policies will likely better support these important behaviors in both childcare settings.

## Data Availability

Data are available from the corresponding author upon request.

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
