# Peer review of "Feeding and Activity Environments for Infants and Toddlers in Childcare Centers and Family Childcare Homes in Southeastern New England"

_ijerph, 2022, doi:10.3390/ijerph19159702_

Round 1

Reviewer 1 Report

Dear Authors,

It is with great interest that I read the manuscript titled Feeding and Activity Environments for Infants and Toddlers in Childcare Centers and Family Childcare Homes, in which you have evaluated the food and activity environments for infants and toddlers in 21 childcare centers and 20 family childcare homes and set out to highlight and analyze differences by childcare type. The article undoubtedly has strengths: it is original and insightful. It deals with a highly relevant rea of childcare research, so it is praiseworthy in that regard. Furthermore, it relies on solid and well explicated methodology, as far as I was able to determine.

The fact that the childcare centers and home accounted for by the authors were all located in the US in the Providence, RI area should be specified in the title. elaborating and expounding upon provider behavior is insightful and provides a relevant perspective and evaluation tool.

I believe a higher degree of contextualization ought to be provided when discussing "healthy foods". That standard is not well elucidated and the phrase is bandied about without providing any discussion as to a set of criteria delineating nutritional values, dietary intake for the children at the centers herein examined and so on. More on how the healthy food standard is defined ought to be provided, in addition to the extent to which practices and dynamics unfolding at care centers may influence the adoption of healthy or unhealthy habits at home. I would also recommend discussing the role of family environments in molding the child's willingness to take up healthy habits. Childcare environments and family environments can in fact both play prominent roles in shaping the child's habits and their likelihood of being overweight or obese later on, but such correlations are not adequately fleshed out by the authors.

I do believe that th article is a worthy endeavor which should be broader in terms of contextualization and discussion. Too much attention on the data and too little on their interpretation, as it stands.

Regardless, I feel the authors deserve praise for their sound analysis and competent delivery.

The article is well written, although a few typos can be found here and there. Further proofreading is advisable. The tables are well conceived and effective at conveying key points of the analytical process.

I look forward to an improved, more well-rounded version of this noteworthy research article.

Sincerely,

Author Response

Thank you for your thoughtful comments. I have attached a table of all revisions based on each of your comments. 

Reviewer 2 Report

Line 100   Describe “trained observer.” Was it one consistent observer? How was the observer trained? What criteria did the observer meet?

Line 115-117 and Line 135 and Line 142-143   Please clarify that the observations were of program activities or individual providers, not of individual children. Also, were more than one provider in a classroom observed and if yes, how was that counted?

Line 340-352 This is a concerning paragraph that seems to imply, based on the cited research, that the expectation is that providers who are Hispanic would use authoritative practices. As you point out, your data do not support findings from other research, but consider if your findings are strong enough to enter this fray. Would you be willing to state that FCCH providers who are Hispanic are less authoritarian in feeding practices? Is it important to have this section when you did not find or in fact include this as a research question? Given the small number of providers and programs in your study, is it necessary to address this any more than you would report practices based on education or age? It seems there is too little data to include this section.

Table 5 and Line 306    Clarify” summary of feeding and active play provided to parents. Are these only written summaries? Is it a written form? Did observers count conversations with families?

Lines 353-357   For those unfamiliar with your instrument, it would be helpful to give examples of “active play” and “moderate play” from the EPAO.

Lines 458-459   This conclusion includes an important statement about your findings that differences exist between child care setting types. Though you rightly indicate more study is needed, consider a paragraph in your discussion that elaborates organization that might influence these practices. You address some of those logistics throughout the discussion, but a summary of those would be helpful.

Author Response

(The authors gave the same response as above.)

Reviewer 3 Report

This paper is well-written and presented. My comments are minimal as seen below:

Please define abbreviations at first mention. E.g PA. (Line 23 of abstract).

What is the aim of the study? This should be clearly stated in the first few lines of the abstract before commenting on the methodology used in the study.

Once an abbreviation is defined at first mention continue to use that abbreviation throughout the manuscript. Do not revert to the full word. E.g Physical Activity in line 67. 

What does RI mean? Line 91.

What does MA mean? Line 95.

State the policy implications of your study.

Author Response

(The authors gave the same response as above.)

Round 2

Reviewer 1 Report

Dear Authors,

I appreciate the effort you have put into improving your article, which I feel now reads even more thorough and well-rounded.

I believe that the article's strengths (its relevance, novelty, comprehensiveness and interest to the journal's readership) make it worthy of publication.

Good luck on your future endeavors.

Sincerely,